# Knowledge, Attitudes and Perceptions about Cervical Cancer Risk, Prevention and Human Papilloma Virus (HPV) in Vulnerable Women in Greece

**DOI:** 10.3390/ijerph17186892

**Published:** 2020-09-21

**Authors:** Elena Riza, Argiro Karakosta, Thomas Tsiampalis, Despoina Lazarou, Angeliki Karachaliou, Spyridon Ntelis, Vasilios Karageorgiou, Theodora Psaltopoulou

**Affiliations:** 1Department of Hygiene, Epidemiology and Medical Statistics, Medical School, National and Kapodistrian University of Athens, 11527 Athens, Greece; arg.karakosta@yahoo.gr (A.K.); thomastsiamp@med.uoa.gr (T.T.); karagela@hotmail.com (A.K.); spydel95@gmail.com (S.N.); vaskarageorg@hotmail.com (V.K.); tpsaltop@med.uoa.gr (T.P.); 2Institute of Human Sciences, Wadham College, University of Oxford, Oxford OX1 3PN, UK; despoina.lazarou@wadham.ox.ac.uk

**Keywords:** cervical cancer risk factors, secondary prevention, screening, Pap smear, HPV, behavioural model, vulnerable populations, refugees, migrants, Roma women

## Abstract

Cervical cancer can be largely preventable through primary and secondary prevention activities. Following the financial crisis in Greece since 2011 and the increased number of refugees/migrants since 2015 the proportion of vulnerable population groups in Greece increased greatly and the ability of the healthcare sector to respond and to cover the health needs of the population is put under tremendous stress. A cross-sectional study was designed to assess the characteristics of vulnerable women in Greece regarding cervical cancer risk factors, prevention through screening activities and Human Papilloma Virus (HPV) knowledge. Two cohorts of women aged 18 to 70 years were studied (142 in 2012 and 122 in 2017) who completed an interviewer-administered questionnaire based on the behavioural model for vulnerable populations. According to this model, the factors that affect the behaviour of women in relation with their knowledge, attitudes and beliefs towards cervical cancer and the HPV vaccine in our study sample are categorised in predisposing factors (age, educational status, nationality menopausal status and housing) and enabling factors (lack of insurance coverage). Results from both univariate and multivariate analyses show that older age, low educational background, refugee/migrant or ethnic minority (Roma) background, menopausal status, housing conditions and lack of insurance coverage are linked with insufficient knowledge on risk factors for cervical cancer and false attitudes and perceptions on cervical cancer preventive activities (Pap smear and HPV vaccine). This is the first study in Greece showing the lack of knowledge and the poor attitudes and perceptions on cervical cancer screening and the HPV vaccine in various groups of vulnerable women. Our results indicate the need of health education and intervention activities according to the characteristics and needs of each group.

## 1. Introduction

Cervical cancer is the fourth most common cancer in women in Europe after cancers of the breast, colon and lung, with 122,000 new cases per year (6.6% of total cancer cases in women) having a great impact on a woman’s life, and being a major cause of mortality (29,600 deaths, 3.5% of total female deaths) [1].

Cervical cancer can be largely preventable through primary and secondary prevention activities. The Human Papilloma Virus (HPV) vaccination for girls has been introduced in 37 of the 53 countries of the WHO European Region as part of their national routine immunisation programmes [2], while boys, in most cases, are not included. However, all vaccinated women should be regularly screened for cervical cancer following national guidelines (e.g., with Pap smear, visual inspection), as the HPV vaccine protects from 71% up to 90% of the HPV types associated with cervical cancer. However, the vaccine does not offer protection from types of HPV already present before vaccination. Infection with several HPV types is very common, and 80% of men and women will become infected at some point in their lives, thereby making HPV the most commonly sexually transmitted disease, especially in the age group of up to 25 years [2].

According to the declaration of Astana in (2015) on Universal Health Coverage (UHC) in Primary Health Care (PHC), all people, including those who are vulnerable or marginalised, should have access to health care services without any administrative or financial restriction. One of the prerequisites of UHC is the health literacy of people, which, if present, allows them to make informed choices about their health, especially where prevention of disease is concerned. One of the challenges stated in the Astana declaration is the empowerment of people and communities to acquire the knowledge, skills and resources to promote their own health, especially the young and the vulnerable. Both HPV vaccination and screening for cervical cancer are health interventions ideally fitted in a primary health care setting [3].

Vulnerability is frequently linked with low health literacy and low adherence to health promotion programmes. Factors linked with vulnerability, especially in women, are poverty, low educational level, unemployment and migrant/refugee or ethnic minority status. Some of these factors are also linked to a high risk of HPV infection and increased cervical cancer prevalence. Knowledge and attitudes towards these issues are of utmost importance in increasing uptake of primary and secondary prevention activities [4].

Since 2011, Greece has been in the midst of a severe financial crisis that has resulted in the imposition of severe austerity measures and structural reforms in the primary healthcare system, along with substantial salary cuts and increasing unemployment rates in all population groups. Moreover, since 2011, and more intensely since 2015, a substantial number of refugees and migrants entered the country as a result of the political unrest in the greater Middle East area and the impoverished status of several African countries [5].

As a consequence, the proportion of vulnerable population groups in Greece increased greatly and the ability of the primary health care sector to respond and to cover the health needs of the population is put under tremendous stress. In view of this unprecedented humanitarian crisis and in order to cover the urgent health needs, several Non-Governmental Organizatiosns (NGOs) operate polyclinics in key areas of Greece addressing vulnerable population groups.

The participation of vulnerable women in prevention activities, such as the primary and secondary prevention of cervical cancer through HPV vaccination and regular administration of Pap smear tests, is usually limited—mostly due to reasons related to social exclusion factors that inherently hinder access to the healthcare services [6,7,8]. Moreover, there are indications that, apart from the socio-economic characteristics of these vulnerable groups, factors related to health behaviour that reflect knowledge level, attitudes and perceptions towards the disease contribute to the low participation of these women. These perceptions usually relate to cultural elements and local customs [6].

It has also been shown that the residence status in a country is directly linked to the level of health service use, with the persons who do not possess legal documents of residence to abstain from any health care provision service [9].

In addition, women from ethnic minorities have steadily lower participation rates in prevention activities compared to the non-minority groups in the same country. For this reason, the behavioural model for vulnerable populations [10] is extensively used to study the factors that affect participation to disease prevention activities.

This model provides the theoretical background to categorise the factors that affect the use of healthcare services in vulnerable population groups that include refugee and migrant women as well as women from ethnic minority groups [11]. This model defines predisposing, enabling and need factors that regulate healthcare service use. Predisposing factors include age, family status, number of children, health beliefs, nationality, social integration level, housing type, number of persons under the same roof, educational level and employment status. Enabling factors reflect the ability of the person to use the healthcare services such as insurance coverage, income, social benefit eligibility and frequency of healthcare service use. Need factors include both direct factors, such as presence of disease, and indirect factors, such as individual perception of health status [10].

The behavioural model for vulnerable populations is an adaptation of the model described by Andersen in 1995 [12], which states that there are factors and social circumstances that affect the predisposition of individuals to use the healthcare service that enable or hinder their need to seek health care.

In Greece, there is minimal research activity regarding the study of factors affecting the participation of women from vulnerable groups in disease prevention programmes and it is mostly directed towards the health of Roma women [13,14]. The only information of healthcare system access regarding refugee and migrant women comes from fragmented non-peer reviewed information in progress reports of NGOs operating in Greece.

To the best of our knowledge, this is the first methodologically structured effort in Greece to identify the knowledge gaps and barriers of women from vulnerable groups that affect their participation in disease prevention programmes, such as Pap smear tests for the prevention of cervical cancer and HPV immunisations.

The aim of the present study is to identify the level of knowledge, the attitudes and perceptions towards cervical cancer prevention and HPV vaccination of women in vulnerable groups in Greece, and to compare the differences among women in vulnerable groups with variable socio-economic characteristics, nationalities and ethnic backgrounds.

## 2. Materials and Methods

### 2.1. Study design and Research Questions

A cross-sectional study design was applied to study research questions relating to (a) the socio-economic characteristics of vulnerable women in Greece who seek healthcare services through NGOs, (b) the uptake of Pap smear testing reflecting the level of participation in secondary prevention activities, (c) the level of knowledge on the causes and risk factors of cervical cancer and (d) the attitudes and perceptions of cervical cancer and the HPV vaccine.

### 2.2. Study Sample and Study Setting

The study participants were selected from the population of women attending the polyclinics run by two NGOs in Greece (Doctors of the World—Medecins du Monde—MdM and PRAKSIS). These NGOs provide primary healthcare services free of charge to vulnerable population groups irrespective of legal status, race, national or ethnic background. Information was collected in two periods (2012 and 2017) in four different polyclinics (two in Athens; one run by MdM and one run by PRAKSIS, one in Perama and one in Patras, both run by MdM). These time periods were selected considering that the year 2012 reflects the impact of the austerity measures following the 2011 financial crisis in Greece and the year 2017 reflects the impact of the increased refugee/migrant influx in the country after the massive displacement of populations in 2015.

Data collection took place during November/December 2012 and May/June 2017, as the patients’ attendance is particularly high during these months according to the NGOs’ clinic attendance records.

### 2.3. Eligibility Criteria

Eligible study participants were women aged 18 to 70 years of age, as they form the target population group for participation in cervical cancer screening programmes, and in most cases these women are responsible for making choices on the HPV vaccination of their children. Five Roma women younger than 18 years (three aged 16 years and two aged 17 years, all married, with one–three children each) were allowed to participate as Roma have, on average, their first child at a much younger age than Greeks (at an age of 14–15 years); hence, they are eligible to participate in cervical cancer screening activities and they form an exceptionally vulnerable group. Consent for participation was given by their partners.

The study participants were selected from the women who attended the polyclinic visiting a gynaecologist or paediatrician during the study period to facilitate data collection. The NGO polyclinics are mostly run by volunteer medical practitioners, and as such, the days and times of operation of all clinics including gynaecology and paediatric are scheduled on a weekly basis, upon availability.

No specific exclusion criteria were applied. However, four women refused to participate, due to lack of linguistic communication and time pressure (one Rom, who spoke only Romani, one Greek, one Bulgarian and one Afghan woman).

### 2.4. Ethical Approval

The NGOs granted permission of access to our team of interviewers to access the population in the polyclinics. The study’s protocol received ethical approval from the bioethics committee of the Medical School, National and Kapodistrian University of Athens (approval number 1718034664 study protocol 14249). All study participants were informed that they participated in a research study on women’s health and were asked to give their consent for participation.

### 2.5. Data Collection

A specially designed questionnaire was created for data collection with closed-ended and scale questions to facilitate completion and coding, based on factors identified through a detailed literature review, according to the elements described in the behavioural model for vulnerable populations [10]. All data were collected anonymously through a personal interview during the time of women’s visit to the gynaecologist or paediatrician in the polyclinic, according to the weekly timetable provided to our team by the NGOs. Our team consisted of five trained interviewers who were present in the polyclinics during the periods of November/December 2012 and May/June 2017.

The questionnaire was divided into three sections: (a) demographic characteristics: age, date of birth, country of origin, mother tongue, number of children in the family, short gynaecological history, level of education, area and type of residence, as well as cohabitation with other people, (b) information in the women’s experience in cervical cancer screening with Pap test and (c) questions to check the knowledge, attitudes and perceptions of women regarding cervical cancer aetiology, HPV infection and HPV vaccination. The questionnaire was administered in Greek, English and French. The questionnaire was pretested in 20 patients, prior to the initiation of the study to check the formulation of the questions and to train the interviewers.

### 2.6. Variables

For the purposes of the present study, as outcome variables, the women’s likelihood of never having a Pap smear test as well as their cognitive score on cervical cancer aetiology and prevention were considered. Two scores of knowledge were calculated, each as the sum of the correct individual questions from the corresponding sections of the questionnaire. More specifically, the cognitive score of cervical cancer in general consists of four questions about the cause of cervical cancer and the area affected, the risk factors for HPV infection and the aim of the HPV vaccine, while the second score consists of 10 questions referring to the Pap test itself, the symptoms of cervical cancer and the HPV vaccine target group. The following factors were used as possible predictors: age, parity, number of children, nationality, level of education, occupation, family size, living conditions, type of residence, number of co-habitants and the duration of residence in Greece.

### 2.7. Study Size and Sampling

Power study estimates in order to obtain a sample response rate of over 85% suggested a minimum required sample size of 120 women. In our study a total of 142 women were interviewed in the 2012 period and 122 women in the 2017 period within the age range 18–70 years using the method of purposive sampling [15,16], to ensure adequate participation from different population groups, namely Greek women, Roma, refugees and migrants. During the preparation and conduct of the pilot study we noticed that the participation rate of Greeks and refugees and migrants in the polyclinics was very similar, while for Roma women participation was significantly smaller. Based on these data, we decided that the 75–85% of the total sample would be composed by Greek and refugee and migrant women while the rest, 15–20%, of Roma, all aged 18 to 70 years old.

### 2.8. Statistical Analysis

For the purposes of the analysis the study participants were allocated into three groups (a) uninsured Greek women, (b) refugee/migrant women and (c) Roma women (Greek and Albanian). Refugee and migrant women were mostly from countries in the Middle East, such as Syria, Afghanistan and Iran, countries in Africa, such as Nigeria, Ethiopia, Cameroon and Kenya and Eastern European countries, such as Albania, Bulgaria and Georgia.

Descriptive statistics are presented in relative frequencies and in mean (standard deviation) form separately for each nationality as well as for the total sample of participants in both study periods. Pearson chi-square test was used for the comparison of the categorical characteristics’ distribution and a Mann-Whitney U test was used for the comparison of continuous measurements between the two study periods in each nationality. Concerning the three main outcomes of interest, generalised linear models [17] were employed. More specifically, concerning the effect of women’s characteristics on the likelihood of never having conducted a Pap test logistic regression analysis was employed, either on univariate or multivariate level. With regards to the women’s scores of knowledge concerning cervical cancer (in general and for disease prevention), Poisson regression analysis was used due to the discrete nature of both outcomes. Finally, after concluding on the logistic and log-linear multivariate regression models the interaction between the nationality and level of education was also examined (non-significant). The SPSS v.24 (IBM Corp, Armonk, NY, USA) and STATA v.13 (StataCorp LP, College Station, TX, USA) software programmes were used for the statistical analysis [18,19].

## 3. Results

The personal, family and residence characteristics of the participating women in both study periods (2012 and 2017), for the total sample and separately for each study group (Greek, Refugees/Migrants and Roma) are shown in Table 1. The mean age of the total sample is approximately 40 years in both study periods. Women in 2017 are more educated (*p* < 0.001) with a larger proportion living in apartments (*p* < 0.001) compared to study participants in 2012. Among women born in Greece, significant differences exist between the two study periods, with those participating in 2017 being older (*p* = 0.009) and having a higher level of education (*p* < 0.001). A smaller percentage is postmenopausal (*p* = 0.013), the vast majority of women have children (96.4%; *p* = 0.013) and approximately eight out of 10 women live in apartments (*p* = 0.001), while these numbers were much smaller in 2012 (deterioration of living standards). In the other two groups (refugees/migrants and Roma women), no significant differences in the distribution of their characteristics between the two study periods are present, except for the fact that fewer Roma women in 2017 have children (*p* = 0.014) and more refugees/migrants live in apartments (*p* = 0.020) compared to 2012 (housing scheme). The recording of smoking habits, a risk factor for the development of cervical cancer, was very low in our study sample, and the variable was not included in the statistical analyses.

### 3.1. Univariate Analyses

#### 3.1.1. Frequency of Pap Smear Testing—Likelihood of Never Having Had a Pap Test

Refugees/migrants and Roma women in 2012 were approximately three times and six times more likely (higher odds) to never having had a Pap test in their lives, respectively (refugees/migrants: OR = 2.93; 95% CI = 1.38–6.22 and Roma: OR = 6.05; 95% CI = 1.82–20.13), compared to Greek women (Table 2). In 2017, Roma women have approximately 40 times higher odds (OR = 40.44; 95% CI = 6.79–240.86) and refugees/migrants approximately seven times higher odds (OR = 7.37; 95% CI = 2.02–26.89) of never having had a Pap test compared to Greek women. In both study periods, younger women and women with a lower level of education (no school or <6 years of schooling) have significantly higher odds of never having had a Pap test. In 2012 postmenopausal women are approximately 50% less likely (OR = 0.47; 95% CI = 0.23–0.96) to never having had a Pap test, compared to the premenopausal ones as well as women living in larger families (>5 persons) (OR = 1.16; 95% CI = 1.01–1.33). Among the study participants in 2017, parous women are 84.0% less likely (OR = 0.16; 95% CI = 0.05–0.48) to never having had a Pap test, compared to the nulliparous ones.

#### 3.1.2. Score of Knowledge about Cervical Cancer Risk Factors

In 2012, the mean number of correct responses on the risk factors and causes of cervical cancer given by Roma women is lower, by 67.0% (IRR = 0.33; 95% CI = 0.12–0.91), and for refugees/migrants is lower, by 59.0% (IRR = 0.41; 95% CI = 0.24–0.69), compared to Greek women (Table 3). The same relationship is also present in 2017, with the difference between Greek and Roma women being even sharper (Roma: IRR = 0.08; 95% CI = 0.01–0.55). In both study periods, a lower educational level (<6 years of schooling) is significantly associated with fewer correct answers, with the mean number of correct answers given by women with a low educational level; 51.0% lower in 2012 (IRR = 0.49; 95% CI = 0.25–0.95) and 55.0% lower in 2017 (IRR = 0.45; 95% CI = 0.25–0.81). Finally, in 2017, postmenopausal women had a lower mean number of correct answers, by 46.0% (IRR = 0.54; 95% CI = 0.33–0.89), compared to premenopausal women. For the 2012 period, there is an indication that women living in apartments have the lowest average number of correct answers compared to those living in detached (independent) houses or other type of houses (Detached house: IRR = 1.90; 95% CI = 1.11–3.26 and Other type of house: IRR = 1.52; 95% CI = 0.78–2.97) (Table 3).

#### 3.1.3. Score of Attitudes and Perceptions on Cervical Cancer Prevention and HPV Vaccination

In both study periods, Roma and refugee/migrant women appear to have incorrect attitudes and perceptions towards the prevention of cervical cancer compared to Greeks, taking into account nationality, level of education and age. Women with a low educational level (<6 years of schooling) give fewer correct answers on the prevention of cervical cancer and HPV vaccination compared to women with a high educational level (>12 years of schooling), whereas the mean number of correct answers seems to increase as the women’s age increases. In 2012, for every additional child in the family, the mean number of correct answers decreases, by 12.0% (IRR = 0.88; 95% CI = 0.83–0.93), the mean number of correct answers given by postmenopausal women is greater, by 22.0% (IRR = 1.22; 95% CI = 1.05–1.42), compared to the premenopausal ones, while women living in detached houses give more correct answers compared to women living in apartments (IRR = 1.11; 95% CI = 0.94–1.31). In 2017, the average number of correct answers given by women with children is higher, by 64.0% (IRR = 1.64; 95% CI = 1.26–2.13), compared to women without children (Table 3).

### 3.2. Multivariate Analyses

#### 3.2.1. Frequency of Pap Smear Testing—Likelihood of Never Having Had a Pap Test

In both study periods, women’s nationality and level of education are significant social determinants for their likelihood of never having had a Pap test. In 2012 the participants’ age is also statistically significantly related, while in 2017, the parity appears to be of borderline significance. After adjusting for the other factors, the three groups of study participants seem to differ significantly, with those born in Greece having the lowest odds of never having had a Pap test (2012: OR = 1.59; 95% CI = 0.40–6.36 (Roma), OR = 3.69; 95% CI = 1.43–9.55 (refugees/migrants) and 2017: OR = 14.13; 95% CI = 1.97–101.27 (Roma), OR = 6.18; 95% CI = 1.60–23.86 (refugees/migrants)). Women with a higher level of education (>12 years of schooling), are more likely to have had a Pap test (2012: OR = 12.03; 95% CI = 2.92–49.54 (Low level), OR = 2.69; 95% CI = 0.75–9.60 (Middle level 7–11 years) and 2017: OR = 5.45; 95% CI = 1.05–28.28 (Low), OR = 1.53; 95% CI = 0.36–6.44 (Middle)) In 2012 older women (OR = 0.95; 95% CI = 0.92–0.98) and in 2017 women without children (OR = 0.32; 95% CI = 0.09–1.14) have higher odds of never having had a Pap test (Table 4).

#### 3.2.2. Score of Knowledge about Cervical Cancer Causes and Risk Factors

In both study periods, women’s nationality and level of education were significant predictors for their score of knowledge about cervical cancer, while in 2012 the type of house was of borderline significance and in 2017, menopausal status was significantly related to the amount of correct responses given by participants. After adjusting for the other factors, in both study periods, the mean number of correct answers given by Roma and refugee/migrant women is much smaller compared to those given by Greek women (2012: IRR = 0.49; 95% CI = 0.17–1.43 (Roma), IRR = 0.30; 95% CI = 0.16–0.57 (refugees/migrants) and 2017: IRR = 0.10; 95% CI = 0.01–0.76 (Roma), IRR = 0.64; 95% CI = 0.45–0.92 (refugees/migrants)), while lower educational level was significantly associated with lower score of knowledge about cervical cancer (2012: IRR = 0.24; 95% CI = 0.11–0.53 (Low), IRR = 0.48; 95% CI = 0.24–0.97 (Middle) and 2017: IRR = 0.63; 95% CI = 0.35–1.14 (Low), IRR = 0.57; 95% CI = 0.39–0.84 (Middle)). In addition, in 2012, compared to women living in apartments, those living in detached (independent) houses had greater knowledge about cervical cancer, since their mean number of correct answers is 74.0% higher (IRR = 1.74; 95% CI = 1.01–3.03). In 2017, the mean knowledge score of postmenopausal women is by 46.0% lower (IRR = 0.54; 95% CI = 0.33–0.89), compared to the premenopausal ones.

#### 3.2.3. Score of Attitudes and Perceptions about Cervical Cancer Prevention

Regarding the women’s attitudes and prevention on cervical cancer prevention, the multivariate analysis showed that nationality and age remained statistically significant predictors for the relevant score in both study periods, whereas in 2012, menopause, level of education and number of children in the family were also significant predictors. More specifically, after adjusting for the other factors, the average score for refugees/migrants in 2012 was statistically significantly lower by 40.0% (IRR = 0.60; 95% CI = 0.49–0.73), and in 2017, the mean score of refugees/migrants and Roma women was significantly lower, by 18.0% (IRR = 0.82; 95% CI = 0.71–0.95) and by 53.0% (IRR = 0.47; 95% CI = 0.32–0.70), respectively, compared to Greek women. Older participants seemed to have a higher score in both periods (2012: IRR = 1.006; 95% CI = 1.000–1.013 and 2017: IRR = 1.008; 95% CI = 1.001–1.015). Finally, in 2012, there was an indication that postmenopausal women had scored higher by 17.0% (IRR = 1.17; 95% CI = 0.98–1.40), while women with more children had significantly fewer correct answers on cervical cancer prevention (IRR = 0.87; 95% CI = 0.82–0.94) (Table 5).

## 4. Discussion

Among the study participants in the two NGO polyclinics in both study periods (2012 and 2017), women born in Greece with over 12 years of education were more likely to have had a Pap test after adjusting for several factors compared to refugee/migrant and Roma women. In addition, older women (in 2012) and nulliparous women (in 2017) were also less likely to conduct a Pap test.

As far as women’s knowledge about cervical cancer in general is concerned, after adjusting for other factors, refugees/migrants and Roma have a lower knowledge profile compared to Greeks with lower educational level being an additional factor both in 2012 and in 2017. In 2012, housing conditions were also significant, in that women living in detached houses (indicating a higher socioeconomic profile) seemed to have greater knowledge compared to women living in apartments (indicating lower socioeconomic profile), and in 2017, postmenopausal women had a lower knowledge score. Knowledge on the HPV vaccine was very low in both study cohorts.

In terms of attitudes and perceptions on cervical cancer prevention, older women and those born in Greece had higher scores in both periods, while in 2012, postmenopausal women, with a higher educational level and those with fewer children in the family (three or less) gave more correct answers.

The level of knowledge on cervical cancer causes and risk factors as well as the identification of attitudes and perceptions on primary and secondary prevention activities, especially in women from vulnerable groups, is a major issue with important implications for public health. As such, it is important to define these characteristics in women with refugee/migrant and ethnic minority backgrounds in order to facilitate their participation in these activities. Moreover, factors like those intensified by the recent global financial crisis, such as changes in the healthcare service provision, especially with regards to cancer prevention, must be taken into account. It is vital to note that women in particular have been gravely affected by the austerity measures imposed mostly through salary cuts, increased poverty, job loss and lack of insurance coverage.

Participation in disease prevention activities is linked to perception awareness of their benefits, which during the years of financial hardship is suppressed or even ignored because of a change in need prioritisation. Moreover, false perceptions on the beneficial effects of regular cervical screening (e.g., by means of a Pap test) and of the HPV vaccination puts many women at higher risk of HPV infection and of cervical cancer, which may result in delayed diagnosis and worse disease outcome.

Evidence from such studies in Greece is very limited mainly because of the opportunistic nature of cervical cancer screening in the country. There is no organised population screening programme to cover the whole of Greece, apart from some efforts that operate at the regional level. The few studies published thus far provide limited and fragmental information from specific population groups such as adolescents, students and health professionals, without specific reference to vulnerability, ethnic minority or migrant or refugee status [20,21,22] In all these studies, a knowledge gap in cervical cancer risk factors and preventive methods has been identified as well as low uptake of Pap smear testing and low HPV vaccination rates. The results of a cross-sectional study on the use of preventive services in a sample of 9682 women aged 40–69 years who participated in an opportunistic regional breast cancer screening programme in Greece showed that 6.6% of the participants never had any Pap test and 71.2% had one Pap test in the previous three years [23]. Older age, low educational status (<6 years of schooling), absence of health insurance and immigrant status were the factors associated with low uptake of preventive services.

A study in Ethiopia of 583 women of childbearing age showed that 46.3% (270 women) had poor knowledge in cervical cancer risk factors and only 9.9% (58 women) had ever been screened with a Pap test [24]. Reasons given were being unaware of the preventive service, fear of discrimination, financial hardship and not having any symptoms.

A qualitative study of 25 women aged 38–62 years from ethnic minority groups in the UK revealed they had a significantly lower participation in the NHS cancer screening preventive services compared to the local population [7]. Reasons mentioned were language difficulties, lack of information, fear of diagnosis and the perception that cancer is a Western disease.

Similar findings were recorded in the qualitative study in groups of women from ethnic minority groups aged 23 to 72 years in a deprived area in Denmark [6] regarding knowledge and attitudes towards cancer prevention. Knowledge on the disease was fragmented and perceptions such as the incurability of cancer and relevance of action only in the presence of symptoms were prevalent.

The results of the present study are in agreement with those from studies conducted in other populations. A major difference is the fact that our sample of women in Greece lack health insurance coverage and that there is no national cancer screening programme where they can get tested.

Age is, as expected, an important factor regarding knowledge and attitudes for cervical cancer, and Roma women have by far lower scores in Pap test participation rates and knowledge about and attitudes towards cervical cancer. These differences indicate the tremendous differences in education and social inclusion levels in Greek and refugee/migrant women. It must be noted that the groups of Greek and refugee/migrant women included in this study were directly comparable in terms of demographic characteristics and the differences lay mainly in duration of stay in the country, integration level and cultural background from the countries of origin.

Research data on Roma women is very limited, as their participation in health-related research presents several methodological difficulties [25]. Roma women in Greece are among the hardest to reach population groups and their participation in health activities is very low. Data for Roma women in our study derives from a small number of participants who attended a polyclinic, a fact which indicates that these women are somehow sensitised towards health issues. The characteristics of the Roma women in our study are compatible with those reported elsewhere in terms of low educational status, poor housing, high fertility rates and social exclusion [26].

Therefore, despite the relatively small number of Roma women in our study, we expect our findings to be even more attenuated in larger cohorts of Roma women where health awareness and probably health literacy levels will be even lower. The same should be expected for the differences between Greek and refugee/migrant women. However, further research in larger groups of these vulnerable populations is necessary to confirm our findings.

In our study, the lack of knowledge and the poor attitudes and perceptions on cervical cancer screening and the HPV vaccine indicate the need of education and intervention, as is the case in many other studies [6,7,23,24]. It is also important to note that there is a high percentage of ignorance of the relationship between HPV infection and cervical cancer as well as the preventive effect of the HPV vaccine, especially in younger ages and mothers of adolescents, despite the fact that a national immunisation programme for HPV has been operating in Greece since 2008.

According to the latest report on HPV in Greece [27], the most frequent types of HPV in Greece are 6, 11, 16, 18 and 33, with 16 being the most prevalent type followed by HPV 18, 31 and 33, which are among the carcinogenic types of the virus. Studies on HPV prevalence in migrant or refugee women are very limited, but studies in Italy and Spain report HPV prevalence of the same types [28,29]. The move of refugees and migrants in Greece is expected to have led to an increased prevalence of the most common HPV types (16 and 18), but studies are required to assess the full spectrum of the HPV types in these populations.

Currently in Greece, the bivalent (HPV types 16, 18) and 9-valent HPV vaccines are available. The HPV vaccine is included in the National Vaccination Programme of the Greek Ministry of Health and its cost is fully reimbursed by the national insurance agency for girls aged 12–16 years. The 9-valent HPV vaccine protects against the infection with HPV types 6, 11, 16, 18, 31, 33, 45, 52 and 58. According to the currently available HPV types prevalence data for Greece, this vaccine should offer adequate protection from the high risk oncogenic types (16, 18) of the virus, provided it is administered at the indicated ages and prior to any infection with these types of the HPV virus.

Based on the currently available data on the prevalence of HPV types in refugee/migrant women, it is expected that both the bivalent and the 9-valent vaccines offer protection against the most prevalent high risk (16,18) types of the virus; therefore, efforts should be made to increase the vaccination rates of these groups.

In addition to the recommendation for girls, in Greece, HPV vaccination is also recommended for specific groups of girls and boys aged 11–26 years (e.g., those with auto-immune diseases, malignant neoplasms, transplanted patients, HIV positives).

Currently no data are available on HPV vaccination rates in men in Greece. The few cross-sectional studies on special male population groups studying the prevalence of sexually transmitted diseases (such as HIV positives, Men who have Sex with Men, partners of women with HPV lesions) have reported HPV prevalence 54.6%–78.4% for anal and 20.3–66.7% for penile cancers (most common HPV types, 6, 11, 16 and 18) [30].

Studies in adolescent boys and girls have shown that there is limited knowledge on the oncogenic properties of HPV and lack of awareness on the HPV vaccine, which is more pronounced in boys [20,31,32]. Further research is required especially in general, low HPV risk male populations and interventions to increase HPV vaccine awareness.

Moreover, a significant percentage of women declared they do not need to have a Pap test across the three different population groups. Many Roma and refugee/migrant women declared they never had a Pap test (48.2 and 70.8% respectively) as well as a Pap test frequency of over two years in the three groups (46.2% in Greeks, 66.7% in refugee/migrants and 53.8% in Roma women). Even the percentages of Pap tests during the past 12 months were smaller (37% in Greeks, 42.9% in Roma and 43.3% in refugees/migrants) compared to the results of similar studies [33,34].

Currently, all individuals holding a national insurance number in Greece may receive vaccinations that are included in the National Vaccination Programme. However, none of these vaccinations are mandatory, thus, the uptake of vaccinations is the responsibility of the local health authorities under the general supervision of the Ministry of Health. It is also important to stress that there is currently no nationally organised cervical cancer screening programme in Greece. Two regional population-based programmes operate covering women aged 18–65 years. The majority of cytology testing in the country is performed upon the decision of the individual to be tested (opportunistic screening). The planning of the Ministry of Health to develop a national population-based cervical cancer screening programme is currently unclear.

### Strengths and Limitations

It is important to note that women in our study sample have already chosen to make contact with the healthcare system in the polyclinics, thus, we can assume that they are more sensitised towards health compared to other women in the same socio-economic level (unemployed and uninsured). This confirms the finding that participation in health prevention activities is higher in the presence of a medical problem [35], and this may have introduced some degree of information bias. As such, our results may not be generalised to the relevant refugee/migrant and Roma groups; however, the effect is probably non-differential.

Another limitation is the self-reported status of the collected data, which in the given circumstances may have introduced some recall bias. Due to the nature of the study, we did not collect any medical history information to cross-check for data accuracy.

Finally, in the groups of refugee/migrant and Roma women, the interviews were not conducted in their native languages, which may have resulted in some communication difficulties.

The strengths of the study include the random nature of the participants from the NGO polyclinics operating in the Greater Athens area and in Patras and the fact that we managed to recruit large numbers of deprived Greek and Roma women who are very hard to reach. This study has brought together vulnerable groups of women from different backgrounds and has managed to make comparisons and find common elements of factors affecting the prevention of cervical cancer and of cancer screening. According to the WHO report on migrant health in Europe, refugee/migrant populations are at higher risk for cervical cancer in the coming years compared to the local populations and actions towards prevention are needed [36].

Overall, we notice that nationality is associated with a differentiation in knowledge and attitudes on cervical cancer aetiology and the HPV vaccine, with Greek women scoring higher than refugee/migrant and Roma women, though they still score quite low. The difference is more attenuated between Roma and Greek and refugee/migrant women, which can be explained given the marginalisation and social exclusion Roma women face. Of interest is the fact that refugee/migrant women who visit the NGO polyclinics seem to be more integrated, as they are in the majority holding legal papers to stay in the country.

Based on the health behavioural model for vulnerable populations, we can categorise the factors that affect the behaviour of women in relation with their knowledge, attitudes and beliefs towards cervical cancer and the HPV vaccine in our study sample in predisposing factors (age, educational status, nationality menopausal status and housing) and enabling factors (lack of insurance coverage).

Organised cancer screening programmes have been criticised for their high cost especially in countries that face financial crisis. However, the health authorities should focus on the effectiveness of these programmes in reducing morbidity and mortality, on facilitating the participation of women from vulnerable groups and on advocating the necessary changes in the healthcare systems [37].

## 5. Conclusions

Our study shows there is need for health education on the prevention of cervical cancer, routine examination with Pap test and HPV vaccination in women from vulnerable groups.

Women also suggested increasing access by way of changes to health care delivery systems and policy, including more direct patient-provider and patient-clinic communications, addressing delays caused by high patient volume, combining preventive services, expanding insurance coverage and adjusting screening guidelines.

As has been shown [38], one of the most powerful factors to increase participation to cancer screening programme is the advice of a health professional who explains the process, the risks and the benefits. In the case of vulnerable women, such an intervention requires consideration of the different cultural, social and religious characteristics along with changes to the healthcare delivery systems.

The decision of a person to act on health issues begins with risk awareness that is facilitated by the health professional who will give advice. We must not forget that in addition to the factors identified in our study, women tend not to prioritise their own health needs because many tend to be nurturers and carers, even when they do have a health problem. Information must take into account the health literacy level and the special characteristics of the population addressed, as in the case of low educated women, refugee/migrant status and social exclusion. Mothers in particular are an important group to which facilitating the participation to HPV immunisation and cervical cancer screening activities is crucial.

## Figures and Tables

**Table 1 ijerph-17-06892-t001:** Personal, family and residence characteristics separately for each study period for the total sample of participating women in the and by nationality.

Variable Group	Variables	Total		*p*	Greek		*p*	Roma		*p*	Other Country/Migrants		*p*
		2012 (*N* = 142)	2017 (*N* = 122)		2012 (*N* = 60)	2017 (*N* = 55)		2012 (*N* = 18)	2017 (*N* = 10)		2012 (*N* = 64)	2017 (*N* = 57)	
Personal characteristics	**Age [years; Mean (sd)]**	40.45 (14.70)	39.34 (11.54)	0.668	46.63 (15.49)	40.71 (9.77)	**0.009 ****	32.44 (12.27)	28.60 (13.30)	0.382	36.91 (12.29)	39.91 (12.01)	0.137
	**Age (%)**												
	≤25 years old	19.00	11.50	**0.005 ****	11.70	10.90	**0.002 ****	33.30	30.00	0.793	21.90	8.80	0.105 *
	26–45 years old	41.50	61.50		36.70	67.30		38.90	30.00		46.90	61.40	
	46+ years old	39.40	27.00		51.70	21.80		27.80	40.00		31.20	29.80	
	**Menopause (% Yes)**	35.90	24.20	**0.040 ****	45.00	22.60	**0.013 ****	16.70	40.00	0.172	32.80	22.80	0.222
	**Education (%)**												
	None/Primary school < 6 y	45.70	19.00	**<0.001 *****	50.00	10.90	**<0.001 *****	93.80	77.80	0.238	29.70	17.50	0.102 *
	High school/Lyceum 7–11 y	38.60	57.00		46.70	61.80		6.20	22.20		39.10	57.90	
	University/Technical school > 12 y	15.70	24.00		3.30	27.30		0.00	0.00		31.20	24.60	
	**Working status** **(% Employed)**	15.70	15.60	0.975	15.00	18.20	0.646	23.50	10.00	0.382	14.30	14.00	0.969
	**Having children** **(% Yes)**	82.40	86.90	0.315	81.70	96.40	**0.013 ****	100.00	70.00	**0.014 ****	78.10	80.70	0.727
Family’s characteristics	**Number of children in the family [Median (Q_1_–Q_3_)]**	2 (2–3)	2 (1–2)	**0.001 ****	2 (2–3)	2 (1–2)	**0.014 ****	4 (2–5)	3 (1–5)	0.238	2 (1–3)	2 (1–2)	0.136
	**Number of people in the family [Median (Q_1_–Q_3_)]**	3 (2–4)	3 (2–4)	0.494	3 (1–4)	3 (2–4)	**0.047 ****	5 (4–10)	4 (2–9)	0.386	3 (2–4)	3 (2–4)	0.939
Residence’s characteristics	**Type of house (%)**												
	Detached house	36.00	24.40	**<0.001 *****	48.30	20.00	**0.001 ****	41.20	80.00	0.301	23.80	17.50	**0.020 ****
	Apartment	44.60	69.90		36.70	74.50		29.40	20.00		55.60	75.50	
	Built house	1.40	0.00		1.70	0.00		5.90	0.00		0.00	0.00	
	Portakabin	3.60	4.10		6.70	5.50		5.90	0.00		0.00	3.50	
	Tent	0.70	0.00		0.00	0.00		0.00	0.00		1.60	0.00	
	Other	13.70	1.60		6.60	0.00		17.60	0.00		19.00	3.50	

The sd symbol represents the standard deviation. Q_1_ and Q_3_ represent the first and the third quartile of the characteristic’s distribution, respectively. *p*-value refers to the comparison of the characteristics’ distribution between the two study periods (*** *p* < 0.001, ** *p* < 0.05, * *p* < 0.10). Concerning the categorical characteristics, the *p*-value is given by the Pearson Chi-square test, while the continuous characteristics is given by the Mann-Whitney U test. Statistically significant values are marked in bold.

**Table 2 ijerph-17-06892-t002:** Odds Ratio, 95% Confidence Intervals and the corresponding level of statistical significance concerning the effect of women’s characteristics on the likelihood they had never undergone a Pap test for each study period.

Models	Variables	Study Period: 2012			Study Period: 2017		
		OR	95% CI	*p*	OR	95% CI	*p*
Model 1	**Nationality**			**0.002 ****			**<0.001 *****
	Greek ^1^	1.00	-	-	1.00	-	-
	Other country/migrants	2.93	[1.38, 6.22]	0.005 **	7.37	[2.02, 26.89]	0.003 **
	Roma	6.05	[1.82, 20.13]	0.003 **	40.44	[6.79, 240.86]	<0.001 ***
Model 2	**Age in years ^2^**	0.95	[0.92, 0.97]	**<0.001 *****	0.96	[0.92, 0.99]	**0.039 ****
Model 3	**Having children**			0.230			**0.001 ****
	No ^1^	1.00	-	-	1.00	-	-
	Yes	1.75	[0.70, 4.39]		0.16	[0.05, 0.48]	
Model 4	**Number of children in the family ^2^**	1.24	[0.98, 1.57]	**0.075 ***	1.06	[0.74, 1.52]	0.762
Model 5	**Menopause**			**0.039 ****			0.376
	No ^1^	1.00	-	-	1.00	-	-
	Yes	0.47	[0.23, 0.96]		1.55	[0.59, 4.05]	
Model 6	**Working status**			0.492			0.472
	Unemployed ^1^	1.00	-	-	1.00	-	-
	Employed	0.71	[0.26, 1.90]		0.62	[0.17, 2.30]	
Model 7	**Type of house**			**0.053 ***			0.402
	Apartment ^1^	1.00	-	-	1.00	-	-
	Detached house	0.65	[0.30, 1.42]	0.281	1.83	[0.71, 4.72]	0.213
	Other	2.22	[0.87, 5.66]	**0.096 ***	0.68	[0.08, 6.00]	0.726
Model 8	**Number of people in the house ^2^**	1.16	[1.01, 1.33]	**0.035 ****	1.16	[0.97, 1.40]	0.106
Model 9	**Level of education**			**0.010 ****			**0.001 ****
	High (University) ^1^	1.00	-	-	1.00	-	-
	Low (None-Primary school)	4.37	[1.44, 13.30]	**0.009 ****	9.46	[2.22, 40.24]	**0.002 ****
	Middle (High school-Lyceum)	1.85	[0.59, 5.79]	0.293	1.64	[0.42, 6.39]	0.473

The results presented are based on logistic regression analysis. ^1^ Baseline category for each characteristic. ^2^ Odds Ratio refers to one unit change of each characteristic. The “Other” type of house includes: built house, prefabricated house, tent and any other possible type of house. In each model (Model 1–Model 9) only one characteristic is included (Univariate analysis). CI = Confidence Interval and OR = Odds Ratio (*** *p* < 0.001, ** *p* < 0.05, * *p* < 0.10). Statistically significant values are marked in bold.

**Table 3 ijerph-17-06892-t003:** Incidence Rate Ratio, 95% Confidence Intervals and the corresponding level of statistical significance concerning the effect of women’s characteristics on the average score regarding their knowledge about the cervical cancer and their knowledge about the prevention of cervical cancer, for each study period.

Models	Variables	Score of Knowledge about Cervical Cancer Risk Factors						Score of Knowledge about the Prevention of Cervical Cancer					
		Study Period: 2012			Study Period: 2017			Study Period: 2012			Study Period: 2017		
		IRR	95% CI	*p*	IRR	95% CI	*p*	IRR	95% CI	*p*	IRR	95% CI	*p*
**Model 1**	**Nationality**			**0.001 ****			**0.005 ****			**<0.001 *****			**<0.001 *****
	Greek ^1^	1.00	-	**-**	1.00	-	**-**	1.00	-	**-**	1.00	-	**-**
	Other country/migrants	0.41	[0.24, 0.69]	**0.001 ****	0.67	[0.47, 0.96]	**0.030 ****	0.66	[0.56, 0.77]	**<0.001 *****	0.82	[0.71, 0.95]	**0.008 ****
	Roma	0.33	[0.12, 0.91]	**0.033 ****	0.08	[0.01, 0.55]	**0.011 ****	0.69	[0.53, 0.90]	**0.005 ****	0.43	[0.29, 0.64]	**<0.001 *****
**Model 2**	**Age in years ^2^**	1.00	[0.99, 1.01]	0.780	1.01	[0.99, 1.03]	0.207	1.007	[1.002, 1.012]	**0.008 ****	1.011	[1.004, 1.017]	**0.001 ****
**Model 3**	**Having children**			0.853			0.260			0.442			**<0.001 *****
	No ^1^	1.00	-	**-**	1.00	-	**-**	1.00	-	**-**	1.00	-	**-**
	Yes	0.95	[0.52, 1.73]		1.43	[0.77, 2.65]		0.93	[0.77, 1.12]		1.64	[1.26, 2.13]	
**Model 4**	**Number of children in the family ^2^**	0.89	[0.74, 1.07]	0.197	0.93	[0.79, 1.08]	0.338	0.88	[0.83, 0.93]	**<0.001 *****	0.95	[0.89, 1.01]	0.118
**Model 5**	**Menopause**			0.244			**0.015 ****			**0.010 ****			0.383
	No ^1^	1.00	-	**-**	1.00	-	**-**	1.00	-	**-**	1.00	-	**-**
	Yes	1.33	[0.83, 2.14]		0.54	[0.33, 0.89]		1.22	[1.05, 1.42]		0.93	[0.78, 1.10]	
**Model 6**	**Working status**			0.399			0.969			0.507			0.114
	Unemployed ^1^	1.00	-	-	1.00	-	-	1.00	-	-	1.00	-	-
	Employed	1.31	[0.70, 2.44]		0.99	[0.61, 1.62]		1.08	[0.87, 1.33]		1.17	[0.96, 1.41]	
**Model 7**	**Type of house**			**0.066 ***			**0.141**			**0.020 ****			0.294
	Apartment ^1^	1.00	-	**-**	1.00	-	**-**	1.00	-	**-**	1.00	-	-
	Detached house	1.90	[1.11, 3.26]	**0.020 ****	0.85	[0.56, 1.31]	**0.463**	1.11	[0.94, 1.31]	**0.211**	0.87	[0.73, 1.04]	0.128
	Other	1.52	[0.78, 2.97]	0.223	0.15	[0.02, 1.09]	**0.061 ***	0.80	[0.64, 1.01]	**0.055 ***	0.92	[0.66, 1.26]	0.588
**Model 8**	**Number of people in the house ^2^**	0.95	[0.86, 1.05]	0.330	0.95	[0.87, 1.05]	0.315	0.96	[0.93, 0.99]	**0.004 ****	0.99	[0.95, 1.02]	0.519
**Model 9**	**Level of education**			**0.047 ****			**0.007 ****			**<0.001 *****			**0.002 ****
	High (University > 12 yrs) ^1^	1.00	-	**-**	1.00	-	**-**	1.00	-	**-**	1.00	-	**-**
	Low (None-Primary school < 6 yrs)	0.49	[0.25, 0.95]	**0.034 ****	0.45	[0.25, 0.81]	**0.007 ****	0.72	[0.58, 0.89]	**0.002 ****	0.65	[0.52, 0.83]	**<0.001 *****
	Middle (High school-Lyceum 7–11 yrs)	0.90	[0.49, 1.65]	0.725	0.60	[0.41, 0.88]	**0.010 ****	0.99	[0.81, 1.22]	0.942	0.89	[0.75, 1.05]	0.162

The score of knowledge about the cervical cancer contains questions that refer to: (a) the point of occurrence, (b) the cause of occurrence, (c) the risk factors and (d) the aim of the vaccine for the Human Papilloma Virus (HPV) virus (four questions). The score of knowledge about the prevention of cervical cancer includes questions referring to test Pap, to the symptoms of cervical cancer and to the vaccine for HPV virus (10 questions). The final scores were calculated as the sum of the right answers given by the participants. The results presented are based on Poisson regression analysis. ^1^ Baseline category for each characteristic. ^2^ Incidence Rate Ratio refers to one unit change of each characteristic. The “Other” type of house includes: built house, prefabricated house, tent and any other possible type of house. In each model (Model 1–Model 9) only one characteristic is included (Univariate analysis). CI = Confidence Interval and IRR = Incidence Rate Ratio (*** *p* < 0.001, ** *p* < 0.05, * *p* < 0.10). Statistically significant values are marked in bold.

**Table 4 ijerph-17-06892-t004:** Results from multivariate logistic regression models concerning the effect of women’s characteristics on the likelihood they had never undergone a Pap test, for each study period. Odds Ratio, 95% Confidence Intervals and the corresponding level of statistical significance.

	Statistically Significant Predictors	OR	95% CI	*p*
Study Period: 2012	**Nationality**			**0.027 ****
	Greek ^1^	1.00	-	**-**
	Other country/Migrants	3.69	[1.43, 9.55]	**0.007 ****
	Roma	1.59	[0.40, 6.36]	0.516
	**Level of education**			**0.001 ****
	High (University) ^1^	1.00	-	**-**
	Low (None-Primary school)	12.03	[2.92, 49.54]	**0.001 ****
	Middle (High school-Lyceum)	2.69	[0.75, 9.60]	0.127
	**Age in years ^2^**	0.95	[0.92, 0.98]	**0.001 ****
Study Period: 2017	**Nationality**			**0.010 ****
	Greek ^1^	1.00	-	**-**
	Other country/Migrants	6.18	[1.60, 23.86]	**0.008 ****
	Roma	14.13	[1.97, 101.27]	**0.008 ****
	**Level of education**			**0.070 ***
	High (University) ^1^	1.00	-	**-**
	Low (None-Primary school)	5.45	[1.05, 28.28]	**0.044 ****
	Middle (High school-Lyceum)	1.53	[0.36, 6.44]	0.561
	**Having children**			**0.079 ***
	No ^1^	1.00	-	**-**
	Yes	0.32	[0.09, 1.14]	

The final models presented in the table are based on the backward selection procedure. After testing the women’s characteristics presented in Table 2, only those with *p*-value less than 0.10 were kept in the final models. **^1^** Baseline category for each characteristic. **^2^** Odds Ratio refers to one unit change in participants’ age. CI = Confidence Interval and OR = Odds Ratio (** *p* < 0.05, * *p* < 0.10). Statistically significant values are marked in bold

**Table 5 ijerph-17-06892-t005:** Results from multivariate Poisson regression concerning the effect of women’s characteristics on the average score regarding their knowledge about the cervical cancer and their knowledge about the prevention of cervical cancer, for each study period. Incidence Rate Ratio, 95% Confidence Intervals and the corresponding level of statistical significance.

Score of Knowledge about Cervical Cancer Risk Factors					Score of Knowledge about the Prevention of Cervical Cancer				
	Statistically Significant Predictors	IRR	95% CI	*p*		Statistically Significant Predictors	IRR	95% CI	*p*
Study Period: 2012	**Nationality**			**0.001 ****	Study Period: 2012	**Nationality**			**<0.001 *****
	Greek ^1^	1.00	-	**-**		Greek ^1^	1.00	-	**-**
	Other country/Migrants	0.30	[0.16, 0.57]	**<0.001 ****		Other country/Migrants	0.60	[0.49, 0.73]	**<0.001 *****
	Roma	0.49	[0.17, 1.43]	0.191		Roma	1.13	[0.82, 1.56]	**0.463**
	**Type of house**			**0.098 ***		**Menopause**			**0.078 ***
	Apartment ^1^	1.00	-	**-**		No ^1^	1.00	-	**-**
	Detached house	1.74	[1.01, 3.03]	**0.049 ****		Yes	1.17	[0.98, 1.40]	
	Other	1.76	[0.90, 3.47]	0.101		**Level of education**			**0.001 ****
	**Level of education**			**0.001 ****		High (University) ^1^	1.00	-	**-**
	High (University) ^1^	1.00	-	**-**		Low (None-Primary school)	0.60	[0.45, 0.80]	**0.001 ****
	Low (None-Primary school)	0.24	[0.11, 0.53]	**<0.001 *****		Middle (High school-Lyceum)	0.84	[0.64, 1.11]	0.211
	Middle (High school-Lyceum)	0.48	[0.24, 0.97]	**0.040 ****		**Age in years ^2^**	1.006	[1.000, 1.013]	**0.065 ***
Study Period: 2017	**Nationality**			**0.007 ****		**Number of children in the family ^2^**	0.87	[0.82, 0.94]	**<0.001 ****
	Greek ^1^	1.00	-	**-**	Study Period: 2017	**Nationality**			**0.001 ****
	Other country/Migrants	0.64	[0.45, 0.92]	**0.017 ****		Greek ^1^	1.00	-	**-**
	Roma	0.10	[0.01, 0.76]	**0.026 ****		Other country/Migrants	0.82	[0.71, 0.95]	**0.009 ****
	**Menopause**			**0.016 ****		Roma	0.47	[0.32, 0.70]	**<0.001 ****
	No ^1^	1.00	-	**-**		**Age in years ^2^**	1.008	[1.001, 1.015]	**0.018 ****
	Yes	0.54	[0.33, 0.89]			**Level of education**			0.297
	**Level of education**			**0.015 ****		High (University) ^1^	1.00	-	**-**
	High (University) ^1^	1.00	-	**-**		Low (None-Primary school)	0.82	[0.64, 1.06]	0.132
	Low (None-Primary school)	0.63	[0.35, 1.14]	0.129		Middle (High school-Lyceum)	0.91	[0.77, 1.08]	0.289
	Middle (High school-Lyceum)	0.57	[0.39, 0.84]	**0.004 ****					

The final models presented in the table are based on the backward selection procedure. After testing the women’s characteristics presented in Table 3, only those with *p*-value less than 0.10 were kept in the final models. The score of knowledge about the cervical cancer contains questions which refer to: (a) the point of occurrence, (b) the cause of occurrence, (c) the risk factors and (d) the aim of the vaccine for the HPV virus (four questions). The score of knowledge about the prevention of cervical cancer includes questions referring to test Pap, to the symptoms of cervical cancer and to the vaccine for HPV virus (10 questions). The final scores were calculated as the sum of the right answers given by the participants. ^1^ Baseline category for each characteristic. ^2^ Incidence Rate Ratio refers to one unit change of each characteristic. CI = Confidence Interval and IRR = Incidence Rate Ratio (*** *p*< 0.001, ** *p* < 0.05, * *p* < 0.10). Statistically significant values are marked in bold.

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
