# Peer review of "Knowledge, Attitudes and Perceptions about Cervical Cancer Risk, Prevention and Human Papilloma Virus (HPV) in Vulnerable Women in Greece"

_ijerph, 2020, doi:10.3390/ijerph17186892_

Round 1

Reviewer 1 Report

This is a very well written manuscript that is accompanied by a large fraction of recently published articles in the literature (5 yrs or less).  I feel that it is deserving of publication. However, the study was powered on the basis of the two time groupings (2012 vs 2017). It appears that findings for the Roma group may be seriously under-powered.  It is possible that Greek vs Other country migrants comparisons are also under-powered.  Commentary on this consideration needs to be added to the paper.  This commentary should satisfy the point that the largest differences apply to the group (Roma) with the smallest numbers.

Author Response

Dear Editors and Reviewers,

many kind thanks for reviewing our submitted manuscript. We have taken all your suggestions very seriously and we are now re-submitting our work having addressed all your comments. All authors have been involved in the current work and have approved all changes/additions.

We hope that you will find our revisions satisfactory.

A detailed presentation of the changes made are indicated in the attached file

With kind regards,

Elena Riza, MPH, MSc, PhD

Reviewer 2 Report

In this manuscript by Riza et al., the authors examined a cross-sectional study to assess the characteristics  of vulnerable women in Greece in relation to cervical cancer risk factors and prevention through HPV screening. This is an important study, as enough research and literature is currently available for educating women worldwide, that could significantly reduce the numbers of  deaths from this devastating cancer.  The study asked some critical questions  regarding women’s perspective on HPV and cervical cancer. The fact that the authors took into account education levels of family members is of significance. The report is well written. However, the major focus is on socio-economic conditions in which these populations live. I would like to see the following information added to the writing where appropriate to add to the flow of discussion.

[1] What prevalent HPV types are present in the Greek population ? How has the move  of migrants and refugees from different parts of the world contributed to the prevalence of different HPV types to the epidemiology in Greece ? In other words, is there a change in the prevalent types of HPV before and after the migration of different populations in Greece?

[2] Are the bi- and quadrivalent HPV vaccines currently available effective in targeting the prevalent HPV types present in the Greek population ?

[3] Is there a need to develop HPV vaccines that are specific for targeting the HPV types in this population ?

[4] What is known about HPV vaccination of men in this population ? What are the current views of men receiving vaccination in this population ?

[5] Does Greece have any government policies for implementing HPV vaccinations in this population ? If so, who enforces these policies?

[6] Is there health care coverage for women to receive HPV vaccinations at the appropriate times and is there screening available for early detection ? If not, are there plans for developing these policies ?

[7] Smoking is a risk factor for cervical cancer development. The authors should include this aspect in their discussion.

Author Response

Dear Editors and Reviewers,

many kind thanks for reviewing our submitted manuscript. We have taken all your suggestions very seriously and we are now re-submitting our work having addressed all your comments. All authors have been involved in the current work and have approved all changes/additions.

We hope that you will find our revisions satisfactory.

A detailed presentation of the changes made are indicated in the attached file.

With kind regards,

Elena Riza, MPH, MSc, PhD

Reviewer 3 Report

Revision of the article:

Determinants, knowledge, attitudes and perceptions about cervical cancer risk, prevention and Human Papilloma Virus (HPV) in vulnerable women in Greece

It is a good article that addresses the characteristics of vulnerable women in Greece regarding cervical cancer risk factors and prevention through screening activities, all based on the health behavioural model for vulnerable populations. This study has confirmed the factors that affect the behaviour of women in relation to their knowledge, attitudes and beliefs towards cervical cancer and the HPV vaccine. The findings are important as they further show that there is need for health education on the prevention of cervical cancer, routine examination with Pap test and HPV vaccination in women from vulnerable groups in Greece.

However, there are some aspects to review prior to publication:

  • It would be important to explain in more depth why you decided to include in this study two cohorts of women of two different periods (2012 and 2017). What information do you think this comparison offers and why do you think it is important?
  • In the eligibility criteria (line 139) you specify that eligible study participants were women aged 18 to 65 years of age, but after in Study size & Sampling (line188) you mentioned that “…and 122 women in the 2017 period within de age range 15-70 years…”. This seems an internal inconsistency of the study. It would be convenient to clarify well what was the age range of the participants in the selection criteria.
  • It is necessary to correct some errors in the References section, both in the referencing of articles and electronic resources. You can check the recommendations in:

https://www.mdpi.com/journal/ijerph/instructions

https://www.mdpi.com/authors/references

I hope that the suggested changes help to improve the quality of the article and that they are well received.

Kind regards

Author Response

(The authors gave the same response as above.)

Round 2

Reviewer 2 Report

The authors have addressed my concerns and added new information in the revised manuscript. Thank you!